# Impact of an Educational Program on Improving Nurses’ Management of Fever: An Experimental Study

**DOI:** 10.3390/healthcare10061135

**Published:** 2022-06-17

**Authors:** Bi-Hung Hsiao, Ya-Ling Tzeng, Kwo-Chen Lee, Shu-Hua Lu, Yun-Ping Lin

**Affiliations:** 1Department of Nursing, Taichung Veterans General Hospital, Taichung 407219, Taiwan; judybihung@yahoo.com.tw; 2School of Nursing, China Medical University, Taichung 406040, Taiwan; tyaling@mail.cmu.edu.tw (Y.-L.T.); rubylee@mail.cmu.edu.tw (K.-C.L.); 3Department of Nursing, China Medical University Hospital, Taichung 404332, Taiwan

**Keywords:** attitude, behavior, education, fever management, knowledge, nurse

## Abstract

Background: Despite a public information campaign “To Break the Myth of Fever”, nurses continued to overtreat fever. This study hypothesized that the campaign lacked the detailed rationale essential to alter nurses’ attitudes and behaviors. Aim: To evaluate the effect of the educational program on nurses’ knowledge, attitudes, and behaviors related to fever management. Design: A randomized experimental design using a time series analysis. Methods: A random sample of 58 medical/surgical nurses was evenly divided into an intervention and a control group. The intervention group received an educational program on fever and fever management. Both groups completed a pretest and four posttests using investigator-developed instruments: a questionnaire on knowledge and attitudes about fever management and a fever treatment checklist to audit charts. Results: The intervention group had markedly higher knowledge scores and reduced use of ice pillows at all four posttests, as well as lower use of antipyretics overall, except for the first posttest, despite no sustained change in attitude. Conclusions: An educational program for fever management can effectively improve clinical nurses’ knowledge and attitudes about fever management.

## 1. Introduction

Fever is a common clinical symptom in up to 36% of patients in general medical wards and up to 50% in critical care [1,2]. Fever is defined as an elevation of body temperature over the normal daily variation associated with the thermoregulatory centers in the hypothalamus when affected by pyrogens [3]. This is the result of controlled phagocytic responses to the pathogen presence and the stimulation of cytokines, which react with arachidonic acid, altering the thermoregulation system of the hypothalamus [4]. In essence, fever is a natural adaptive physiological response to threats. The temperature increase from fever is self-limited [5], and body temperature is typically <41 °C; this type of moderate fever promotes immunity, and fever reduction is redundant because the elevated temperature activates the body’s immune system and restricts the growth of bacteria and viruses [6,7,8,9,10,11,12]. Fever differs from hyperthermia resulting from heatstroke and heat exhaustion, which is not derived from an elevation in the hypothalamic temperature setpoint; instead, it is due to failed thermoregulation resulting in a body temperature greater than 40 °C [8,10,13,14].

Based on our literature review, no evidence supports routine fever reduction for patients with a moderate fever (i.e., temperature below 38.5–39 °C), unless the patients are experiencing cardiopulmonary overload due to the body temperature increase [9]. A meta-analysis demonstrated that there was not a significant difference in the survival time between patients who received more active fever management and those who did not (hazard ratio: 0.91, 95% CI: 0.75–1.10, *p* = 0.32) [15]. Reportedly, fever reduction treatments (e.g., antipyretics, physical cooling, or both) do not shorten the disease progress but, perhaps, prolong it instead [16]. Some studies demonstrated that fever reduction could be harmful [9,13,17]. Fever reduction is unnecessary for patients, except for cases of hypermetabolism due to fever that might aggravate the disease [8,18]. Factors that may influence decisions about administering antipyretics include knowledge, attitudes, and beliefs. The well-recognized benefits regarding fever are documented in the literature; however, this is not the same in clinical practice [19].

Either administrating antipyretics or applying physical cooling (e.g., ice pillows, tepid sponge baths, taking off clothes/quilts, and lowering the air-conditioning temperature) can decrease body temperature/suppress fever. Although giving antipyretics reduces the patient’s temperature and alleviates discomfort, it does not cure the disease; it might cause hepatotoxicity, nephrotoxicity, and gastrointestinal upset, as well as conceal the symptoms of the disease and even impede the diagnosis and treatment of the disease [5,20]. While applying physical cooling can lower body surface temperature by heat conduction, it cannot actually reset the setpoint and might easily cause cold, shivering, and discomfort in patients [5,20,21,22].

Although evidence suggests that fever is beneficial for patients and that moderate fever should not be decreased [9,17,23,24], current nursing practice does not comply with the conclusions of empirical studies. Nurses often try to decrease the body temperature of patients with moderate fever [25,26]. They continue to use the old, outdated practice of using aggressive antipyretics for fever reduction [27,28]. This may be due to a lack of guidance for nurses on how to work better with adults [19]. A survey reported that nurses whose fever/antipyretic knowledge score was lower had a significant negative attitude toward fever (*p* = 0.001) and a significant positive attitude toward antipyretics (*p* < 0.001). In addition, nurses with more professional experiences had a significant positive attitude toward antipyretics (*p* = 0.002) [29]. This could be related to nurses lacking sufficient knowledge about fever management [30,31,32]. Thus, many investigators have used education courses to promote the nursing quality of care for patients with fever [26,33,34,35,36].

The majority of current fever management studies focus on educating pediatric and emergency department nurses in managing fever in the pediatric population [26,33,34,35,36], most of which compared the impact of education on nurses’ knowledge and attitude pre- and post-intervention [26,33,34]. The results showed that fever knowledge improved significantly [34,36,37], and attitudes were found to be more positive [36]. However, most studies used a one-group pretest–posttest design, which did not compare nurses who received an intervention with those who did not [34,37]. In addition, only limited studies reported the changes in nurses’ behavior, such as providing antipyretics to fever patients and/or long-term follow-ups [36]. The study found that one month after the intervention, the experimental group administered antipyretics to patients with an average temperature of 38.73 °C, which was higher than that of the control group (38.26 °C). However, four months after the intervention, the mean temperature at antipyretic administration in the experimental group (38.38 °C) was not significantly different from that in the control group (38.42 °C) [36]. Changes in nurses’ behavior in providing physical cooling, commonly used in clinical practice, have not been explored yet.

Hence, using a two-group experimental design with repeated measures, this study aimed to investigate the immediate, short-term, and long-term effects of an educational program on the knowledge, attitudes, and behaviors of nurses’ fever management for adult patients.

## 2. Methods

### 2.1. Study Design

We employed a randomized experimental design [38] in which the experimental group received an educational program and the control group did not. The research design was used to examine the impact of an educational program on improving fever management among nurses in general medical and surgical wards using a time series analysis. Surveys and medical record audits were used to collect data at five time points: baseline, immediately after intervention completion, and 1, 2, and 3 months posttest.

### 2.2. Ethical Consideration

This study adhered to the protocol approved by the Institutional Review Board of Taichung Veterans General Hospital, Taiwan (IRB TCVGH: CE13247); all participants fully understood the objectives and methods of this study and provided written informed consent before the study.

### 2.3. Study Setting

A random sample was selected from registered nurses in Taichung Veterans General Hospital. The general medical and surgical wards for adults were selected, and an even assignment was performed, in which each of the six wards suitable for recruiting cases was written on a piece of paper, which was then enclosed in an envelope. Through random sampling, the first one was assigned to the intervention group, and the second one was assigned to the control group.

### 2.4. Study Participants

The inclusion criteria were: (1) Level N–N4 nurses. In Taiwan, N–N1 nurses are responsible for basic nursing, N2 are responsible for critical care nursing, N3 are responsible for education and holistic nursing, and N4 are responsible for research and specialized nursing [39]. (2) Nurses who provide clinical care for patients. The exclusion criteria were: (1) part-time nurses, (2) student nurses, and (3) nurses planning to take leave from the study ward for >1 week or relocate to another ward during the 3-month data collection period.

### 2.5. Sample Size

The sample size was finalized per Edwards et al. [36]. The effect size = (μ_1_ − μ_2_)/σ. The knowledge part was (14.49 − 12.38)/2.54 = 0.83. Thus, the effect size was set at 0.8, *α* = 0.05, and power = 0.8. Using G*Power statistical software 3.1 (Heinrich-Heine-Universität, Düsseldorf, Germany), we estimated that 26 participants were needed in each group. Estimating the drop-out rate at 20%, each group recruited 31 participants, with 62 in total.

### 2.6. Intervention

The intervention was an educational program for fever management. To provide nurses with the correct knowledge of fever and fever management, this educational program was based on the contemporary literature and the results of empirical research [5,9,10,11,16,17,18,20,23,28,40,41,42,43,44,45]. The educational materials were developed by the research team and then reviewed by infectious disease physicians.

This program comprised 2 one-hour sessions administered on 2 consecutive days to a group of medical-surgical nurses. The one-hour session included a 40 min lecture and a 20 min discussion led by two researchers (Bi-Hung Hsiao and Shu-Hua Lu). A pretest was conducted before the session, and an immediate posttest was conducted after the second session.

The first session was focused on the knowledge of fever, including: (1) thermoregulation, (2) the definition of fever, (3) causes of fever, (4) mechanisms of fever, (5) stages of fever, (6) benefits and adverse effects of fever, (7) the difference between fever and hyperthermia, and (8) related empirical research results. To reinforce changes in attitudes and behaviors, following the educational session, a group discussion was conducted. The discussion topics involved: (1) Is fever a disease or a symptom? (2) Is a fever the same as hyperthermia? (3) Is fever a friend or an enemy? (4) Does a fever require aggressive body temperature reduction?

The second session was mainly focused on fever management, including: (1) mechanisms of antipyretics and physical cooling, (2) indications of antipyretics and physical cooling, (3) advantages and disadvantages of antipyretics and physical cooling, (4) related empirical research on fever management, and (5) evidence-based recommendations for fever management. After the educational session, a group discussion was again used to strengthen changes in attitudes and behaviors towards fever management. The discussion topics were: (1) Fever is bad! Should the fever be reduced as quickly as possible? (2) Can a prolonged fever potentially cause brain damage? (3) Can an ice pillow really lower the temperature? and (4) Can antipyretics be used for any kind of fever?

### 2.7. Instruments

We used the self-constructed Knowledge and Attitudes about Fever Management Questionnaire (KAFMQ) and Medical Records for Fever Management Checklist (MRFMC) as research tools.

#### 2.7.1. KAFMQ

Content validity: The KAFMQ included basic information, knowledge about fever management (61 items), and attitudes about fever management (12 items). We invited six experts on the content validity index (CVI) to review and judge the questionnaire draft. After deleting inappropriate items, the preliminary questionnaire was completed, which included 57 knowledge items and 10 attitudes items. Knowledge items were made of three elements: physiology of fever, physical cooling, and antipyretics. The CVI values of the knowledge subscales were 0.94, 0.94, and 0.96, respectively, while the CVI value for the attitude scale was 0.99.

Item analysis: we conducted a pilot study on 30 nurses other than research participants and analyzed the difficulty and discrimination for knowledge items. Questions with difficulty <0.1, difficulty >0.8, and discrimination <0.3 were deleted [46]. Then, homogeneity tests analyzed the attitude scale. Items with product–moment correlation coefficients <0.3 were deleted.

After item analyses, the final questionnaire comprised 44 knowledge items and 8 attitude items. The score range for the knowledge scale is between 0 and 44. A higher score indicates good knowledge. The attitude scale was rated on a 4-point Likert scale ranging from 1 (strongly disagree) to 4 (strongly agree) with a score range of 8–32. An overall higher score represents a more positive attitude.

Reliability: the values of the test–retest reliability for the physiology of fever, physical cooling, and antipyretics in the knowledge scale of this final questionnaire were 0.812, 0.723, and 0.656, respectively. For the attitude scale, the Cronbach α coefficient was 0.820.

#### 2.7.2. MRFMC

The MRFMC was primarily used to quantify nurses’ behaviors in using ice pillows and antipyretics. The checklist had two parts: basic information about patients with fever and fever management by nurses; it recorded fever management by nurses to lower the patients’ body temperature during the fever. Before starting the study, 12 patients with fever were enrolled to test the completeness of the checklist. After amending the checklist content, the formal MRFMC was completed.

### 2.8. Data Collection

Data were collected from both groups at baseline, immediately post-intervention, and 1, 2, and 3 months post-intervention. In addition to the KAFMQ, the MRFMC was also performed at each time point to assess medical records within 2 weeks.

### 2.9. Data Analysis

First, descriptive statistical parameters, including frequency, percentage, mean, and standard deviation (SD), were computed. Second, to ensure homogeneity, demographic data were compared between the intervention and control groups using independent-sample t-tests and χ^2^ tests. Third, independent-sample t-tests were used to examine the differences between the group means of the knowledge about and attitude toward fever management. Generalized estimating equations (GEEs) with exchangeable correlation structures were used to test the effects of the group assignments (intervention vs. control), the effects of time (pretest vs. posttest), and the group-by-time interaction for the outcome measure (KAFMQ). Finally, χ^2^ tests were used to detect differences between group means in using ice pillows and antipyretics (MRFMC). A two-tailed *p* value < 0.05 was considered statistically significant. All analyses were conducted using IBM1 SPSS1 Statistics 20 (IBM Corp, Armonk, NY, USA).

## 3. Results

### 3.1. Basic Participant Characteristics

Data were collected from December 2014 to May 2015. In the intervention group, we excluded two participants who did not attend educational courses. In the control group, one participant retired before completing all questionnaires, and another did not finish the 3-month posttest questionnaire. Altogether, this resulted in a total of 58 nurses (intervention group: 29; control group: 29) participating in this study and a drop-out rate of 6.45%. Between these two groups, no significant difference was observed in gender, age, education, nursing ladder, and clinical experience (*p >* 0.05) (Table 1).

### 3.2. Changes in Knowledge Post-Education

Before the educational intervention, the mean score of knowledge items was 27.26 (correction rate: 61.95%) for the intervention group and 26.35 (correction rate: 59.89%) for the control group. We observed no significant difference between the two groups (*p* = 0.473). The results revealed that after the educational intervention, the knowledge scores in the intervention group were significantly higher than the control group at the end of the intervention and 1, 2, and 3 months post-intervention (Table 2).

Based on the GEE results, the differences in knowledge between the baseline and the four post-intervention follow-up tests in the intervention group were more significant than their corresponding differences in the control group (all *Ps* < 0.05). That is, after educational intervention, knowledge about fever management exerted a significant influence at the end of the intervention and 1, 2, and 3 months post-intervention (*β* = 10.68, 95% CI: 8.10–13.26; *β* = 4.43, 95% CI: 1.85–7.01; *β* = 5.04, 95% CI: 2.44–7.63; *β* = 2.75, 95% CI: 0.14–5.36, respectively) (Table 3).

### 3.3. Changes in Attitudes Post-Education

Prior to the educational intervention, no significant difference was noted between the intervention and control groups in attitude scores (21.77 vs. 20.77; *p* = 0.350, respectively). The results revealed that after the intervention, the scores of attitudes in the intervention group were significantly higher than those of the control group at the end of the intervention and 1 month post-intervention. However, at 2 and 3 months post-intervention, these differences were not statistically significant (Table 2).

Based on the GEE results, a significant between-group difference was observed in attitudes at the end of the intervention (*p* < 0.001). That is, after the educational intervention, the attitudes about fever management exerted a significant influence at the end of the intervention, but not 1, 2, and 3 months post-intervention (*β* = 1.83, 95% CI: −0.26 to 3.92; *β* = 0.53, 95% CI: −1.57 to 2.63; *β* = 0.18, 95% CI: −1.92 to 2.29, respectively) (Table 3).

### 3.4. Changes in Behaviors Post-Education

We assessed the effects of the educational intervention on the nurses’ behaviors in providing ice pillows and antipyretics using *χ*^2^ tests. The results revealed that after the educational intervention, the rates of providing ice pillows in the intervention group were lower than the corresponding rates in the control group at the end of the intervention and 1, 2, and 3 months post-intervention. In addition, all of these differences were statistically significant (26.72% vs. 55.41%, *p* < 0.001; 21.25% vs. 68.18%, *p* < 0.001; 41.90% vs. 59.40%, *p* < 0.001; 24.00% vs. 74.38%, *p* < 0.001, respectively) (Figure 1).

As for providing antipyretics, the difference between the intervention group and the control group at the end of the intervention was not significant (13.36% vs. 20.38%, *p* = 0.065). However, the rates of providing antipyretics in the intervention group were lower than the corresponding rates in the control group 1, 2, and 3 months post-intervention. Furthermore, these differences were significant (13.33% vs. 24.68%, *p* = 0.004; 13.97% vs. 30.77%, *p* < 0.001; 11.33% vs. 29.75%, *p* < 0.001, respectively) (Figure 1).

## 4. Discussion

This study used a randomized experimental design of two-group time-series measurements to examine the effects of educational intervention on nurses’ knowledge, attitudes, and behaviors in fever management at baseline, immediately post-intervention, and 1, 2, and 3 months post-intervention. The findings suggest that an educational program can effectively improve nurses’ knowledge, attitudes, and behaviors for adult fever management.

### 4.1. The Effects of Education on Nurses’ Knowledge in Fever Management

The results demonstrate that an educational program could improve the knowledge of nurses for fever management. The scores of knowledge in fever management were increased at 1, 2, and 3 months after the educational intervention. Consistent with previous studies [26,31,34,47], the correction rate (ranging from 59.89% to 61.95%) of fever knowledge items in both groups at baseline was low, suggesting that most clinical nurses did not have satisfactory evidence-based knowledge about fever management and needed further education.

Corroborating previous studies, this study establishes that education can promote nurses’ knowledge of fever management. Khalifa indicated that nurses’ knowledge about the physiology of child fever, the method of measuring body temperature, fever management, administering antipyretics, and document recording were remarkably promoted 1 week and 1 month after education [26]. Likewise, Considine and Brennan similarly concluded that the average number of correctly answered questions increased 2 weeks after two 30 min education courses [34]. Edwards et al. reported that the overall score of knowledge in the intervention group was significantly higher 1 and 4 months after education, proving that education can effectively promote nurses’ knowledge of fever management [36].

### 4.2. Effects of Education on Nurses’ Attitudes in Fever Management

This study demonstrates that education could transform the attitudes of nurses about fever management immediately after education, but the effects did not last, corroborating some previous studies. Considine and Brennan [34] reported that 2 weeks after evidence-based education training, more pediatric nurses agreed that: (1) a child’s body temperature is not relevant to disease seriousness; (2) a child’s body temperature <41 °C might not cause damage; (3) a child’s body temperature is not the only indicator for administering antipyretics; and (4) many nurses displayed a fever phobia attitude.

In contrast, our findings contradict those of Edwards et al. [36], in which changes in nurses’ attitudes toward fever management not only remained 1 month after education but lasted until 4 months after education. Edwards et al. claimed that education promoted knowledge about fever management in the experimental group and, thus, transformed the attitude of the participants in the experimental group [36].

Although our study demonstrated that knowledge was promoted after education, Walsh et al. suggested that high scores of knowledge did not always affect attitudes [31]. Comparing the teaching strategies and research designs between the current study and Edwards et al.’s [36], multiple reasons were noted to elucidate why changes in nurses’ attitudes toward fever management did not last in this study. First, the frequency and length of the courses in our study might not be adequate to continuously affect the attitudes of the intervention group. Second, the lack of peer influence or role modeling in this study could be a factor because our research was conducted on individuals rather than on interpersonal influence or on the whole organization. Third, in addition to in-class teaching, multiple teaching and learning strategies should have been used. Fourth, misconceptions about fever have led to fever phobia [18]. A previous study found that 56.8% of nurses had fever phobia [31]. Furthermore, most nurses are influenced by traditional knowledge and clinical experiences and practices passed down by senior nurses. Additionally, since they usually fear senior nurses with authority, they will just choose outdated practices and attitudes. Overall, this study illustrates that education could transform nurses’ attitudes toward fever management, but the effects did not last. To sustain the effect, besides promoting knowledge, it is suggested that the design of educational courses refer to Edwards et al. [36], in which courses were held at longer intervals, the frequency of courses was increased, professional personnel were assigned to wards to provide counseling, posters were posted, booklets were provided, peer influence was emphasized, and correct ideas were discussed.

### 4.3. Effects of Education on Nurses’ Behaviors in Fever Management

To the best of our knowledge, this is the first study to investigate the effects of educational programs on changing nurses’ behaviors using physical and pharmacy measures of fever reduction. The results established that education could transform the behavior of nurses toward fever management. After education, the rates of using ice pillows and antipyretics were decreased 1, 2, and 3 months post-intervention.

We found that providing ice pillows was the most applied management for patients with fever. Before the educational intervention, more than 50% of patients with fever were managed with ice pillows; this rate declined in the intervention group but not in the control group. Possible explanations for high ice pillow usage in managing patients with fever are: (1) giving ice pillows to someone with fever is a general practice and independent nursing function; (2) ice pillows are easy to obtain, and no physician order is required prior to giving ice pillows to patients; (3) behaviors are easily affected by external pressure from coworkers, family members, and patients; and (4) old knowledge failed to be updated.

However, the empirical research suggests that physical cooling might violate the theory of body temperature regulation and could temporarily lower the body surface temperature rather than resetting the temperature, resulting in shivering and discomfort [21,48]. Hence, Carey suggested that physical cooling not be applied if no antipyretic has been applied to lower the body temperature [16]. Similarly, this study also confirms that education can decrease the use of ice pillows by nurses; it is worth promoting this education.

In addition, this study showed that education could decrease the rates of administering antipyretics, which lasted 3 months after the intervention; this finding differed from Edwards et al. [36], in which the transformed behavior did not continue. The difference in results might be due to different experimental designs. In our study, we measured the antipyretic usage directly, while Edwards et al. [36] used the average temperature of patients receiving antipyretics as an indicator of nurses’ behavior. Despite the differences, both studies demonstrated that promoting nurses’ knowledge could improve the overuse of fever management and reduce the inappropriate usage of antipyretics.

### 4.4. Study Strengths and Limitations

The strengths of this study include the following: (1) This study used a randomized pretest–posttest control group design, which is different from previous studies that used a one-group pretest–posttest design. (2) The educational sessions included both lectures and discussions, and a group discussion was designed to effectively shape attitudes and behaviors towards fever management. (3) Longitudinal follow-ups were used in this study to understand short-term and long-term effects. (4) This study was the first to actually check the use of ice pillows and antipyretic drugs as specific behavioral change indicators.

The present study has several limitations. First, this research was conducted at a medical center in Taiwan, and the conclusions might not be applicable to other medical institutions. Second, based on anonymity, we did not collect the names of participating nurses as recorded in the medical records. Therefore, we only analyzed fever management for the whole medical institution in a cross-sectional manner and could not examine the effects of education on individuals, unlike the analyses of knowledge and attitude, which were conducted longitudinally. Hence, all of the limitations mentioned above should be considered for future research.

## 5. Conclusions and Recommendations

This study demonstrates that an educational program for fever management can effectively improve clinical nurses’ knowledge and attitudes toward fever management. Regarding nurses’ behavior, education can decrease the inappropriate use of ice pillows and antipyretics. Although fever is a common symptom in clinical practice, most clinical nurses do not have adequate knowledge about fever management, and relevant education courses on clinical fever management are rarely provided to them.

Therefore, this study suggests that medical institutions regularly provide continuing education courses on fever management to promote evidence-based fever management and then reduce the number of unnecessary or excessive fever management treatments. However, to promote successful behavior change, a supportive environment is also needed for changing obsolete knowledge, attitudes, and behaviors, such as posting reminder signs and regular revisions of fever management guidelines. Furthermore, providing behavioral role modeling of effective fever management in clinical practice may influence their peers more effectively.

## Figures and Tables

**Figure 1 healthcare-10-01135-f001:**
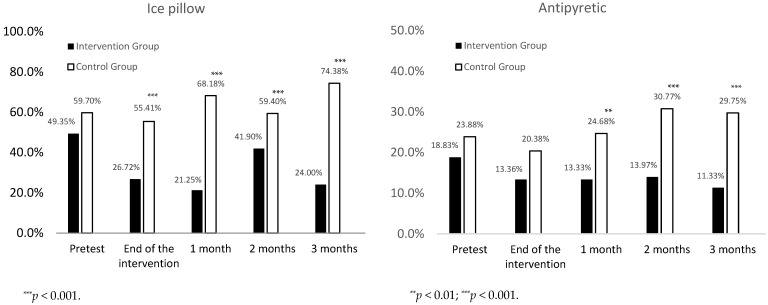
Percentage of ice pillows and antipyretics usage at different time points.

**Table 1 healthcare-10-01135-t001:** Participants’ characteristics.

Variable	Intervention Group (*n* = 29)		Control Group (*n* = 29)		*p*
	*n* (%)	M ± SD	*n* (%)	M ± SD	
Gender					
Female	28 (96.55)		28 (96.55)		>0.999 ^a^
Male	1 (03.44)		1 (03.44)		
Age (years)		27.58 ± 6.77		30.39 ± 8.05	0.143 ^b^
20–24	10 (34.48)		8 (27.58)		0.404 ^a^
25–29	14 (48.27)		11 (37.93)		
30–34	2 (06.89)		2 (06.89)		
≥35	3 (10.34)		8 (27.58)		
Education					
Associate	5 (17.24)		4 (13.79)		0.731 ^a^
College	23 (79.31)		25 (86.20)		
Graduate (or above)	1 (03.44)		0 (00.00)		
Nursing Ladder					
N	7 (24.13)		7 (24.13)		0.241 ^a^
N1	8 (27.58)		3 (10.34)		
N2	12 (41.37)		14 (48.27)		
N3	2 (06.89)		3 (10.34)		
N4	0 (00.00)		2 (06.89)		
Clinical Experience (months)		67.48 ± 89.15		96.48 ± 98.74	0.230 ^b^
0–12	3 (10.34)		4 (13.79)		0.344 ^a^
13–60	17 (58.62)		13 (44.82)		
61–120	6 (20.68)		6 (20.68)		
121–360	3 (10.34)		6 (20.68)		

Note. M, mean; SD, standard deviation; ^a^ Fisher’s exact test; ^b^ *t*-test.

**Table 2 healthcare-10-01135-t002:** Comparison of knowledge and attitude scores between two groups at different time points.

Variable	Intervention Group (*n* = 29)		Control Group (*n* = 29)		*t*	*p*
	Mean	SD	Mean	SD		
Knowledge						
Pretest	27.26	3.89	26.35	5.78	0.722	0.473
End of the intervention	36.93	3.09	25.32	5.61	10.015	<0.001 ***
1 month	34.55	3.47	29.19	4.92	4.847	<0.001 ***
2 months	34.03	2.99	28.03	4.00	6.502	<0.001 ***
3 months	32.62	4.00	29.00	3.63	3.610	0.001 **
Attitude						
Pretest	21.77	3.28	20.77	4.91	0.943	0.350
End of the intervention	25.48	3.44	20.13	3.60	5.879	<0.001 ***
1 month	24.21	3.90	21.35	3.03	3.174	0.002 **
2 months	22.48	5.40	20.90	3.03	1.381	0.174
3 months	22.66	4.46	21.52	2.63	1.184	0.243

Note. ** *p* < 0.01; *** *p* < 0.001.

**Table 3 healthcare-10-01135-t003:** Analysis of changes in knowledge and attitudes after the educational intervention via GEE.

Variable	Knowledge Score	Attitude Score
	(*β* and 95% CI)	(*β* and 95% CI)
Intercept	26.35 (24.85, 27.86) ***	20.77 (19.42, 21.13) ***
Group ^a^ (Experimental vs. Control)	0.90 (−1.22, 3.03)	1.00 (−0.92, 2.92)
Time ^a^		
End of the intervention vs. Pretest	−1.03 (−2.84, 0.77)	−0.65 (−2.10, 0.81)
1 month vs. Pretest	2.84 (1.03, 4.64) **	0.58 (−0.88, 2.04)
2 months vs. Pretest	1.71 (−0.11, 3.54)	0.15 (−1.32, 1.63)
3 months vs. Pretest	2.59 (0.75, 4.43) **	0.67 (−0.81, 2.16)
Group × Time ^a^		
End of the intervention	10.68 (8.10, 13.26) ***	4.33 (2.24, 6.42) ***
1 month	4.43 (1.85, 7.01) **	1.83 (−0.26, 3.92)
2 months	5.04 (2.44, 7.63) ***	0.53 (−1.57, 2.63)
3 months	2.75 (0.14, 5.36) *	0.18 (−1.92, 2.29)

Note. *β* = treatment effect compared with baseline values and/or control group; 95% CI, 95% confidence interval. ^a^ Reference: control group for group effect; baseline values for time effect; and baseline values of control group for interactions; * *p* < 0.05; ** *p* < 0.01; *** *p* < 0.001.

## Data Availability

The data presented in this study are available on request from the corresponding author. The data are not publicly available due to privacy.

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
