# Peer review of "Impact of an Educational Program on Improving Nurses’ Management of Fever: An Experimental Study"

_healthcare, 2022, doi:10.3390/healthcare10061135_

Round 1
Reviewer 1 Report
Dear authors, lease see the following comments and suggestions which I believe would improve the overall quality of the study.
· The title, abstract and keywords are representative. However, I suggest using a fifth keyword as many journals use 5 keywords.
· The study well documents and justifies that the current nursing practice does not comply with the conclusions of empirical studies.
· The authors can augment the related studies section and synthesis the findings reported studies so that readers can create a cognitive background to better interpret the findings of the study.
· Please properly cite the methodological standpoints (e.g., quasi-experimental research design).
· Authors can provide more discussions regarding the effects of education on nurses’ attitude in fever management. Though education has an impact, a focus regarding “role modeling” other nurses can be discussed. Besides, continuity of using traditional approaches (e.g., ice pillows) and the reasons why some nurses still use the obsolete approaches can be elaborated.
· The study is clearly reporting results of empirical research. However, the discussion section can be improved by discussion different perspectives and querying the reasons behind using current and obsolete approaches. Besides, suggestion/implications for future research directions can be expanded.
Author Response
Response to Reviewer 1 Comments
Point 1: The title, abstract, and keywords are representative. However, I suggest using a fifth keyword as many journals use 5 keywords.
Response 1: Thank you for the reviewer's suggestion. We have revised the keywords, using a sixth keyword: attitude; behavior; education; fever management; knowledge; nurse
Point 2: The study well documents and justifies that the current nursing practice does not comply with the conclusions of empirical studies.
Response 2: Thank you for the reviewer’s comments.
Point 3: The authors can augment the related studies section and synthesis the findings reported studies so that readers can create a cognitive background to better interpret the findings of the study.
Response 3: We have provided the information in the revised introduction section. Please see the revised manuscript on page 2, lines 79-89.
Point 4: Please properly cite the methodological standpoints (e.g., quasi-experimental research design)
Response 4: We have added the information in the revised Methods section. Please see the revised manuscript on page 3, lines 97-100.
Point 5: Authors can provide more discussions regarding the effects of education on nurses’ attitudes in fever management. Though education has an impact, a focus regarding “role modeling” other nurses can be discussed. Besides, continuity of using traditional approaches (e.g., ice pillows) and the reasons why some nurses still use the obsolete approaches can be elaborated.
Response 5:
Response to (1) Authors can provide more discussions regarding the effects of education on nurses’ attitude in fever management.
We have added the information in the revised Discussion section. Please see the revised manuscript on pages 8- 9, lines 318-322.
Response to (2) Though education has an impact, a focus regarding “role modeling” for other nurses can be discussed.
We have added the information in the revised Discussion and Conclusions and Recommendations section. Please see the revised manuscript on page 8, line 315; page 10, 387-391.
Response to (3) Besides, continuity of using traditional approaches (e.g., ice pillows) and the reasons why some nurses still use the obsolete approaches can be elaborated.
We have added the information in the revised Discussion section. Please see the revised manuscript on page 9, lines 340-343.
Point 6: The study is clearly reporting the results of empirical research. However, the discussion section can be improved by discussion different perspectives and querying the reasons behind using current and obsolete approaches. Besides, suggestion/implications for future research directions can be expanded.
Response to (1) The study is clearly reporting the results of empirical research. However, the discussion section can be improved by discussing different perspectives and querying the reasons behind using current and obsolete approaches
We have added this information in the revised Discussion section. Please see revised manuscript pages 8-9, line 318-322 and page 9, line 340-343.
Response to (2) Besides, suggestion/implications for future research directions can be expanded.
We have added the information in the revised Conclusions and Recommendations section. Please see the revised manuscript on pages 10, lines 387-391.

Reviewer 2 Report
The manuscript provides an important contribution to the academic community by investigating how educational programs improve nurses’ management skills of fever. The presentation of the article is generally good, and Introduction and research method sections are sound. The discussion is supported by relevant previous studies. I suggest that a few areas may be improved with minor corrections.
The most interesting part of this study is the intervention part. So I recommend the authors make their intervention part more informative so that future practitioners can replicate the education program.
In lines 249, 250, 252, need to be corrected.
In line 272, needs to be corrected.
In line 278, needs to improve readability.
Extend the conclusion section.
Environmental factors may contribute to the results. So if possible, add descriptions of environmental factors in which the nurses worked.
Author Response
Response to Reviewer 2 Comments
Point 1: The manuscript provides an important contribution to the academic community by investigating how educational programs improve nurses’ management skills of fever. The presentation of the article is generally good, and Introduction and research method sections are sound. The discussion is supported by relevant previous studies. I suggest that a few areas may be improved with minor corrections.
Response 1: Thank you for the reviewer’s comments.
Point 2: The most interesting part of this study is the intervention part. So I recommend the authors make their intervention part more informative so that future practitioners can replicate the education program.
Response 2: We have provided the information in the revised Methods section. Please see the revised manuscript on pages 3-4, lines 129-155.
Point 3: In lines 249, 250, 252, need to be corrected.
Response 3: We have made revisions to the above mentioned lines (see line 281).
Point 4: In line 272, needs to be corrected.
Response 4: We have made revisions to line 303.
Point 5: In line 278, needs to improve readability.
Response 5: We have made revisions to the original line 278.
Point 6: Extend the conclusion section.
Response 6: We have provided the information in the revised Conclusion section. Please see the revised manuscript on page 10, lines 387-391.
Point 7: Environmental factors may contribute to the results. So if possible, add descriptions of environmental factors in which the nurses worked.
Response 7: We have provided the information in the revised Conclusion section. Please see the revised manuscript on page 10, lines 387-389.
